# Streak Tube-Based LiDAR for 3D Imaging

**DOI:** 10.3390/s25175348

**Published:** 2025-08-28

**Authors:** Houzhi Cai, Zeng Ye, Fangding Yao, Chao Lv, Xiaohan Cheng, Lijuan Xiang

**Affiliations:** Key Laboratory of Optoelectronic Devices and Systems of Education and Guangdong Province, Shenzhen Key Laboratory of Photonics and Biophotonics, College of Physics and Optoelectronic Engineering, Shenzhen University, Shenzhen 518060, China; hzcai@szu.edu.cn (H.C.); 2310455031@email.szu.edu.cn (Z.Y.); 2310455016@email.szu.edu.cn (F.Y.); 2400233023@mails.szu.edu.cn (C.L.); 2400233033@mails.szu.edu.cn (X.C.)

**Keywords:** streak tube lidar (STIL), ultrafast imaging, streak tube, three-dimensional (3D) reconstruction

## Abstract

Streak cameras, essential for ultrahigh temporal resolution diagnostics in laser-driven inertial confinement fusion, underpin the streak tube imaging LiDAR (STIL) system—a flash LiDAR technology offering high spatiotemporal resolution, precise ranging, enhanced sensitivity, and wide field of view. This study establishes a theoretical model of the STIL system, with numerical simulations predicting limits of temporal and spatial resolutions of ~6 ps and 22.8 lp/mm, respectively. Dynamic simulations of laser backscatter signals from targets at varying depths demonstrate an optimal distance reconstruction accuracy of 98%. An experimental STIL platform was developed, with the key parameters calibrated as follows: scanning speed (16.78 ps/pixel), temporal resolution (14.47 ps), and central cathode spatial resolution (20 lp/mm). The system achieved target imaging through streak camera detection of azimuth-resolved intensity profiles, generating raw streak images. Feature extraction and neural network-based three-dimensional (3D) reconstruction algorithms enabled target reconstruction from the time-of-flight data of short laser pulses, achieving a minimum distance reconstruction error of 3.57%. Experimental results validate the capability of the system to detect fast, low-intensity optical signals while acquiring target range information, ultimately achieving high-frame-rate, high-resolution 3D imaging. These advancements position STIL technology as a promising solution for applications that require micron-scale depth discrimination under dynamic conditions.

## 1. Introduction

A streak camera is an ultrafast diagnostic instrument that offers high temporal resolution at the nanosecond, picosecond, or even femtosecond level, along with micrometer-scale spatial resolution. This instrument plays a critical role in major national scientific research projects and large-scale scientific facilities, such as high-energy physics and laser inertial confinement fusion [1,2]. However, with the advancement and application of streak cameras, their utility is no longer limited to conventional ultrafast laser imaging diagnostics. They have also been successfully applied in other emerging imaging technologies, including streak tube imaging lidar (STIL) technology [3,4,5,6]. As a novel three-dimensional (3D) imaging technique, laser radar three-dimensional imaging relies on precise time-of-flight (ToF) measurements of laser pulses to determine the distance to target objects [7,8,9,10,11].

Streak tube imaging LiDAR technology originated in the late 1990s, and compared to conventional laser radar imaging technologies based on avalanche photodiode arrays (APDs) [12], intensified charge-coupled devices (ICCDs) [13], and single-photon detectors (SPADs) [14], streak cameras demonstrate significantly higher temporal resolution capable of detecting picosecond-scale ultrafast phenomena and also provide advantages such as a wide field of view and high detection sensitivity, making them highly desirable for target detection and maritime security applications [15]. In 1988, S. Williamson first proposed a streak tube-based lidar system and demonstrated its capability to capture one-dimensional spatial intensity profiles of target objects [16]. In 1989, F. K. Knight et al. designed a ranging system employing a streak tube as the detector, enabling the imaging of distinct regions on a target [17]. By 1999, John W. McLean achieved 3D underwater imaging using a streak tube lidar system, facilitating its application in marine monitoring [18]. In 2000, Arete Associates validated the feasibility of an airborne Multiple-slit Streak Tube Imaging Lidar (MS-STIL) system for autonomous target acquisition and recognition imaging in missile seekers through the 3D imaging of ground targets at ultra-long ranges [19]. In 2016, Northrop Grumman Corporation (West Falls Church, VA, USA) developed and manufactured an Airborne Laser Mine Detection System (ALMDS) based on streak tube lidar technology, confirming its capacity for rapid scanning and long-range precise localization [20]. Most recently in 2022, Fugro (Leidschendam, The Netherlands), in collaboration with manufacturer Areté Associates (Los Angeles, CA, USA), designed a Rapid Airborne Multibeam Mapping System (RAMMS) utilizing a streak tube receiver [21,22]; this system provides high-quality coastal and nearshore bathymetric datasets rapidly and cost-effectively, and has been widely adopted for topographic and bathymetric mapping. In China, research institutions, including the Harbin Institute of Technology, the Xi’an Institute of Optics and Precision Mechanics of the Chinese Academy of Sciences, and Shenzhen University, have conducted studies on streak tube-based laser radar imaging technology. In 2010, Wei Jingsong’s team at the Harbin Institute of Technology improved the spatial resolution of their streak tube laser radar system through beam compression and successfully imaged building targets at a distance of 700 m [23]. That same year, Guo Baoping’s group at Shenzhen University used a picosecond streak camera with a temporal resolution higher than 2 ps and nonlinearity of less than 2% as the receiver for their streak tube laser radar system research, achieving a ranging accuracy of 96% [24]. In 2019, Tian Jinshou’s team at the Xi’an Institute of Optics and Precision Mechanics developed an ultra-compact streak tube with a large detection area for laser radar applications and theoretically analyzed the imaging performance of this streak tube laser radar system, demonstrating a range resolution of 10 cm and an azimuth field of view of 56 mrad at an operational range of 2.5 km [25,26], providing important theoretical support and reference for domestic research on streak tube laser radar systems.

This study investigates a streak tube-based laser radar imaging system, performing line-scan imaging of static objects to acquire raw streak images at different object positions. On the basis of the spatiotemporal conversion characteristics of streak cameras and the operational principles of streak tube laser radar systems [27], three-dimensional reconstruction was conducted from the original streak images.

## 2. Streak Tube-Based LiDAR Imaging System

Streak tube LiDAR is an active imaging system composed of a pulsed laser transmitter and a time-resolved streak tube-based backward detection receiver.

As illustrated in Figure 1, the pulsed laser beam is collimated into a fan-shaped profile parallel to the target surface through an optical system. Photons carrying target information are focused by the optical system onto the slit at the front end of the streak camera cathode, where the diagnostic pulse undergoes one-dimensional spatial sampling and is imaged onto the photocathode surface to generate photoelectrons. The emitted electrons are accelerated by a uniform electric field within the grid mesh behind the photocathode and then enter deflection plates driven by a linear time-varying high voltage. The time-dependent deflection forces spatially encode the temporal information of the electron beam. The electron beam is subsequently focused by lenses in the drift region, amplified by a microchannel plate (MCP) image intensifier, and finally strikes the phosphor screen to form a visible spatiotemporal image. A delay synchronization circuit ensures the simultaneous operation of the scanning deflection system of the streak tube and the charge-coupled device (CCD)-based image acquisition system, achieving precise synchronization of the optoelectronic signals. The CCD captures dynamically scanned images from the streak camera, which are processed via a three-dimensional (3D) reconstruction algorithm to reconstruct the three-dimensional profile of the target. By leveraging streak tube scanning imaging, high-resolution grayscale images are obtained. Post-processing of the grayscale data combined with 3D reconstruction significantly enhances the distance resolution of the streak tube imaging LiDAR (STIL) system, enabling high-fidelity reconstruction of the information of the target object.

The surface morphology variations at different positions of the target object cause differences in the arrival time of return echoes at the slit. In the streak image, the grayscale information along the one-dimensional direction parallel to the slit represents the intensity profile of the target, whereas the perpendicular direction corresponds to time-resolved channels created by electron deflection, which records the temporal information of laser echoes from surface features at varying distances.

By employing the original peak detection method to extract the strongest point in the reflected optical signal, both the intensity and distance information of the corresponding target point can be derived from the grayscale and positional information of this point. The relationship between the peak pixel and its position can be expressed as $f\left(i,j\right)$, while the intensity information of the peak point is denoted as $I\left(p,j\right)$, where p represents the frame number of the streak image, and i,j indicate the row and column numbers in the p-th frame streak image. The point with the maximum grayscale value on a single time-resolution channel is selected as the peak point. Consequently, the target intensity image and range image can be expressed as(1)$$I\left(p,j\right)=max\left\{f\left(i,j\right)\right\}$$(2)$$R\left(p,j\right)=i_{max}$$

By extracting the intensity and distance information of the maximum grayscale point in each column of every frame and arranging the extracted feature points in a specific order, we can obtain the intensity image and range image of the target object acquired by the streak camera during this row scan. However, the peak detection method inherently selects local maxima of echo signals as feature points for each column, making it susceptible to interference when noise levels are excessive. Additionally, without considering the shape model of the target object, this approach may fail to process spatially continuous surface variations, resulting in limited reconstruction accuracy. Therefore, we introduce a feedforward neural network-based 3D reconstruction method to reduce local errors.

The core of the feedforward neural network-based 3D reconstruction computational model lies in its forward propagation calculation of outputs, followed by weight parameter adjustments through loss function backpropagation, enabling iterative network re-prediction to enhance target reconstruction accuracy. Information in this neural network architecture flows unidirectionally from the input to the output layers, with multiple hidden layers optionally implemented for iterative computation. Building upon the original peak detection method, we developed an optimized algorithm that establishes a 10-hidden-layer neural network based on the streak tube LiDAR imaging model. The workflow diagram of the neural network-based 3D reconstruction algorithm is illustrated in Figure 2.

As illustrated in Figure 2, the training dataset comprises signals from individual time-resolved channels of multiple captured streak raw images. A rapid training function is implemented, and the Levenberg–Marquardt (LM) algorithm—tailored for nonlinear least-squares optimization—is introduced to the network. This algorithm synergizes the advantages of the gradient descent and Gauss–Newton methods, adaptively adjusting the iteration direction and step size to minimize the error between neural network outputs and expected values while reducing the loss function. The loss function for the STIL system’s raw image processing training model is defined as(3)$$L={\left(S-F\left(S\right)\right)}^{2}+\lambda \cdot val\left(M\right)$$
where ***S*** represents the temporal channel signals from five adjacent columns of input streak images, $F\left(S\right)$ denotes the Gaussian-fitted output signals from the neural network model, ***M*** is the Gaussian mean vector of the five adjacent column signals, and ***λ*** serves as a weighting factor to balance the two loss components.

The neural network is trained using signal values from individual temporal channels of the acquired raw streak images as the training dataset. To enhance reconstruction fidelity by mitigating localized errors, adjacent five-column channels within each temporal resolution are sampled as training inputs for neural network fitting, thereby yielding robust peak predictions for subsequent single time channels. The neural network model is optimized through an optimizer and loss function to minimize errors. Ultimately, this enables training across multiple streak image frames, extracting Gaussian-fitted signals for each column and outputting stable peak values to achieve 3D reconstruction.

To investigate the detection performance of the streak tube LiDAR system in measuring the relative distance information of target objects, we simulated the entire process of electron beam focusing, deflection, scanning, and imaging within the streak camera. The simulation also incorporated information acquisition from different depths of the target object while evaluating key performance metrics that affect the imaging quality of the system.

The dynamic temporal resolution of a streak camera is defined as the minimum time interval between two distinguishable pulses achievable by the streak tube. In simulations, multiple ultrashort pulse pairs with fixed Gaussian-distributed temporal separations are numerically generated and projected onto the photocathode. The electron beam trajectories are projected onto the X-Z plane, producing two electron beam spots on the imaging screen. Simulation tests characterizing the temporal and spatial resolution of the streak tube yield the results presented in Figure 3.

A histogram of electron counts along the *X*-axis is generated and fitted with Gaussian curves, as shown in Figure 3a. The temporal resolution limit is quantified via the Rayleigh criterion; when the minimum probability between the two overlapping beam spots along the *X*-axis reaches 0.69 (below the Rayleigh threshold of 0.707), the streak tube is deemed capable of resolving pulses separated by 5.8 ps. Further reducing the pulse separation to 5.7 ps increases the overlap between pulses. The calculated minimum probability at the valley of the *X*-axis distribution curve increases to 0.71, exceeding the Rayleigh criterion threshold and indicating unresolvable temporal discrimination.

Simulations of electron beams emitted at the central position and off-axis distances of 0.1 mm and 5 mm were conducted, with the electron spot dispersion radii calculated at the imaging plane. The relationship between the off-axis distance and spatial resolution contrast is plotted in Figure 3b. The results demonstrate that as the emission position shifts from the center to 5 mm off-axis, the limit of spatial resolution decreases from 22.8 lp/mm (line pairs per millimeter) to 15.9 lp/mm. The electrons emitted closer to the cathode center exhibit smaller spot radii at the imaging plane, corresponding to higher spatial resolution under identical contrast conditions. In summary, the streak camera achieves a limit of temporal resolution of approximately 6 ps and a limit of spatial resolution of 22.8 lp/mm.

The temporal separation of laser pulses is controlled by adjusting the laser emission interval to regulate the arrival time differences at distinct target surfaces, while electron trajectories are tracked under predefined field strength distributions. The simulated target model consists of a four-tier stepped structure, as illustrated in Figure 4a, with intersurface distances of 5 mm (A–B), 10 mm (B–C), and 15 mm (C–D). Streak data acquisition is performed on three stepped surfaces at varying depths within the same spatial orientation. The streak camera is configured with a tube length of 424 mm and an electron pulse full width at half maximum (FWHM) of 1 ps. Pulse intervals are determined by the specific distances between target surfaces. Three sampling points are selected on each stepped surface to represent its distance information. Sequential data acquisition is conducted from shallow to deep surfaces, and the simulation results are shown in Figure 4.

Analysis of data points acquired from each stepped surface reveals that Figure 4b–d present the electron impact distributions of reflected signals from stepped surfaces A and B, where the horizontal centroids of acquisition points at different positions on each stepped surface are identical, sequentially measured at −0.25 cm, −0.13 cm, and 0 cm; taking Figure 4b as an example, the mean vertical centroid for acquisition points on surface A is 0.32 cm, while the corresponding mean for surface B is 0.84 cm, yielding an imaged distance of 5.2 mm between surfaces A and B; similarly, distance reconstruction applied to the electron impact distributions of surfaces B–C (Figure 4c) and C–D (Figure 4d) obtains imaged distances of 5.2 mm (A–B), 10.5 mm (B–C), and 15.3 mm (C–D); processing coordinates from electron impact distributions through our neural network-based 3D reconstruction algorithm generates the simulated single-frame image of the target object model shown in Figure 4e, which captures the object’s contour along the scanning direction while exhibiting unsmooth artifacts in planar regions due to noise; comparative analysis between theoretical model distances and reconstructed imaging distances yields the distance comparison curve in Figure 4f, demonstrating distance reconstruction accuracies of 96% (A–B), 95% (B–C), and 98% (C–D), thereby confirming effective relative distance detection and validating the feasibility of employing streak cameras as core detection components in LiDAR systems.

## 3. Experimental System and Measurement Results

The scanning speed of the streak camera was calibrated, where higher scanning speeds enable the temporal separation of shorter-interval signals after passing through the scanning circuit, thereby indicating the superior temporal resolution of the system.

The precise distance measured by the Fabry–Pérot etalon enables the determination of the temporal interval (∆t) between adjacent fringes in the camera-acquired interferogram, thereby facilitating the calculation of the scanning speed of the streak camera. In the experimental configuration, the etalon length was designed to be 1.17 m, yielding a measured temporal separation of 5.64 ns between adjacent optical pulse fringes. The total optical path delay was established at 116.5 ns, with precise synchronization achieved between the trigger signal and the laser pulse arrival at the streak camera. Assuming linearity in the camera’s scanning velocity, the resulting spatial separation between adjacent output pulses should remain uniform. Nonlinearity analysis across different scanning speed regimes was conducted via the computational formulations for scan nonlinearity expressed in Equations (4) and (5)(4)$$S=\frac{\delta }{\bar{x}}*100\%$$(5)$$\delta =\sqrt{\frac{1}{n(n-1)}\sum_{i=1}^{n}\left(x_{i}-\bar{x}\right)}$$
where S represents the scan nonlinearity, $x_{i}$ denotes the peak-to-peak spacing between adjacent fringes, $\bar{x}$ indicates the arithmetic mean of inter-fringe peak spacing, and δ characterizes the deviation of individual spacing measurements from the arithmetic mean value.

The experimental setup incorporated a streak camera with six scanning circuit speed ranges, and parametric characterization was conducted on the first four high-speed ranges designated S1, S2, S3, and S4. Through oscilloscopic measurements, the full-frame scanning duration and intrinsic temporal delay for each speed range were systematically recorded. Subsequent computational analysis of scan nonlinearity across different operational ranges was performed, and the quantitative results are rigorously tabulated in Table 1.

As shown by the data in Table 1, higher scanning speed ranges (i.e., larger speed range settings) exhibit increased nonlinearity in the scanning circuitry, which significantly degrades both imaging quality and reconstruction accuracy in streak tube LiDAR systems. To achieve optimal target reconstruction fidelity, speed range 3 (S3) was selected for subsequent streak tube LiDAR experiments. The resulting scanning image captured by the camera is shown in Figure 5a, and the corresponding scanning speed (∆*v*_3_) in this speed range was measured to be 16.78 ps/pixel.

The dynamic temporal resolution of the system was measured using an optical system based on a Fabry–Pérot etalon. For the dynamic testing experiment, a Fabry–Pérot etalon with a length of 0.33 m (generating a temporal interval of 1.1 ns) was used, and the fastest scanning speed range (S1) of the scanning circuit was selected for the temporal resolution measurement. Analysis of the acquired streak image revealed a peak-to-peak spacing of 190 pixels between adjacent streaks, with an average streak width of ∆v = 2.5 pixels, ultimately achieving a system temporal resolution of *τ* = 14.47 ps.

The static spatial resolution of the streak tube without a scanning voltage was measured via a resolution test target. As shown in Figure 5b, the test pattern observed on the phosphor screen exhibited progressively denser line pairs toward the photocathode center, gradually becoming sparser toward the edges. The measured spacings were 0.025 mm at the center, 0.05 mm at the left periphery, and 0.033 mm at the right periphery, resulting in a static spatial resolution of 20 lp/mm at the photocathode center and more than 10 lp/mm at the edge regions.

The streak tube LiDAR imaging system was established on the basis of streak camera calibration data. A stepped target constructed from Polyvinyl Chloride (PVC) material served as the test object (Figure 6a), exhibiting inter-surface distances of 5 cm (A–B), 10 cm (B–C), and 15 cm (C–D) sequentially from right to left. Figure 6b schematically depicts the expanded pulsed laser beam illuminating the target during experimentation, with the laser operating at 532 nm wavelength, 10 Hz repetition rate, and maximum pulse energy of 20 mJ. A scanning lift stage with a vertical displacement range of 130 mm to 680 mm was employed to control the target’s vertical motion during the 22 s dynamic push-broom scanning process. Synchronization between optical signals and electrical triggering was achieved using a DG535 digital delay generator, ensuring sub-200 ps temporal alignment throughout the acquisition cycle. A cylindrical optical lens with a diameter of 25.4 mm was used as the beam expander. During the experiments, the slit width was set to 0.2 mm. The CCD detector featured a pixel size of 5.86 μm and a maximum resolution of 1936 × 1216 pixels.

The experiment utilized the third speed range (S3) of the streak camera scanning circuit, with a full-frame scanning time of 20.4 ns, a scanning speed of 16.78 ps/pixel, scanning nonlinearity of 4.2%, and a range resolution of 4.28 cm. Under dynamic scanning conditions, the stepped target was push-broom scanned, yielding a total of 38 raw streak images. Figure 6d displays a representative single-frame raw image acquired from a single laser pulse scan of the target.

As shown in Figure 6d, the laser pulses reflected from four steps at different distances are sequentially scanned and imaged on the phosphor screen before being captured by the CCD. In the raw streak image, the horizontal dimension (along the slit direction) represents the spatial arrangement of the target object, whereas the vertical dimension (scanning direction) records the temporal channel information. Owing to the Gaussian distribution characteristics and finite pulse width of the laser itself, the streaks exhibit a certain width when the echo signals pass through the scanning circuitry of the streak camera. Multiple frames of raw streak images can be acquired when performing push-broom scanning across different regions of the target object.

The neural network-based 3D reconstruction algorithm effectively restores multi-frame raw streak images, as demonstrated by the reconstructed stepped target profile in Figure 7a. The coordinate system is defined as follows: the *X*-axis aligns with the slit direction, encoding surface dimensions and reflectivity; the *Y*-axis corresponds to the laser’s push-broom scanning velocity and temporal parameters; and the *Z*-axis captures depth information between adjacent steps. The quantitative evaluation reveals significant performance improvements—the neural network reduces the average reconstruction error from 10.33% (using conventional peak detection) to 3.57% through noise-robust fitting algorithms. By leveraging adjacent column echo correlations during training, the algorithm suppresses sharp noise artifacts and minimizes local errors, achieving a 65.4% enhancement in global accuracy compared to pixelwise processing. These results validate the algorithm’s ability to resolve micron-scale depth variations in complex LiDAR scenarios.

## 4. Conclusions

Theoretical and experimental investigations of a streak tube imaging LiDAR (STIL) system are presented in this paper. A theoretical model of the STIL system was developed, with simulations revealing limits of temporal and spatial resolutions of approximately 6 ps and 22.8 lp/mm, respectively. Dynamic simulations of laser-reflected signals from targets at varying depths within the same spatial orientation demonstrate an optimal distance reconstruction accuracy of 98%. Building upon theoretical foundations, an experimental STIL platform was implemented. The following key parameters were calibrated: scanning speed (∆v = 16.78 ps/pixel), temporal resolution (τ = 14.47 ps), and central cathode spatial resolution (20 lp/mm). Multiple raw streak images of test objects were acquired and processed via a neural network-based 3D reconstruction algorithm, achieving a reconstruction error of 3.57%. Experimental data demonstrate the system’s ability for high-precision 3D imaging and reconstruction using streak tube technology, validating its potential for applications requiring micron-scale depth resolution.

## Figures and Tables

**Figure 1 sensors-25-05348-f001:**
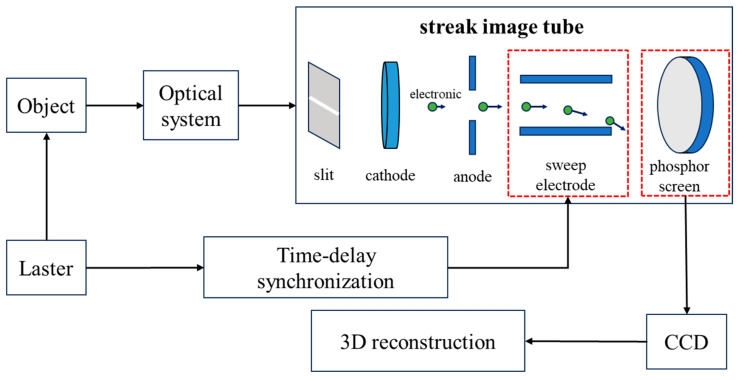
Schematic diagram of the streak tube LiDAR system architecture.

**Figure 2 sensors-25-05348-f002:**
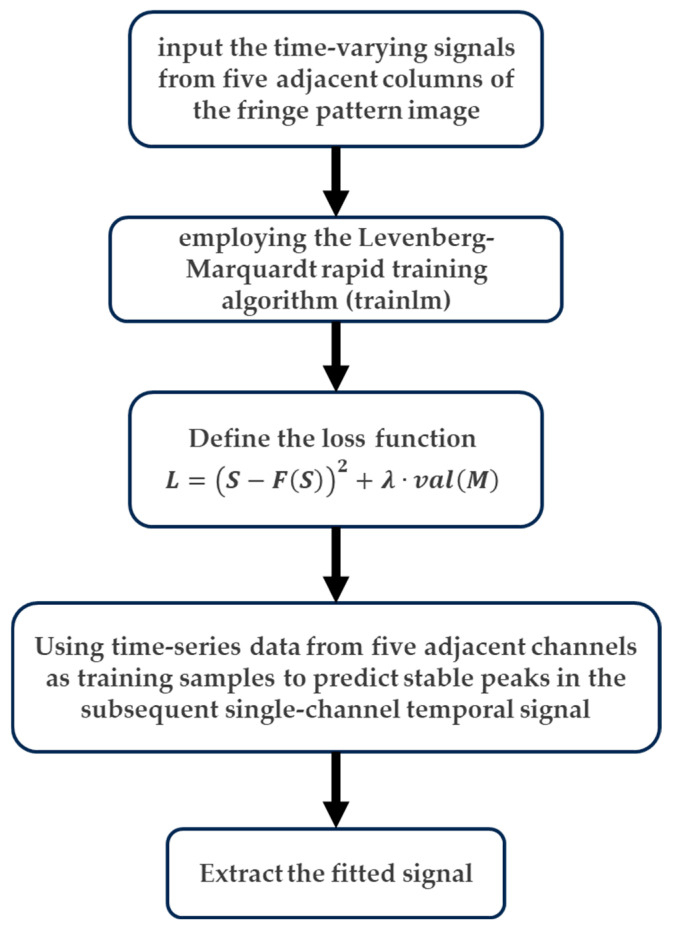
Flow diagram of the neural network-based 3D reconstruction algorithm.

**Figure 3 sensors-25-05348-f003:**
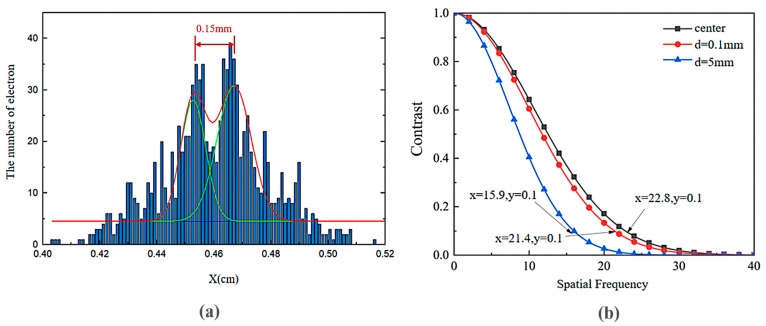
(**a**) Temporal resolution characterization using the Rayleigh criterion; (**b**) spatial resolution at the central position, and off-axis distances of 0.1 mm and 5 mm.

**Figure 4 sensors-25-05348-f004:**
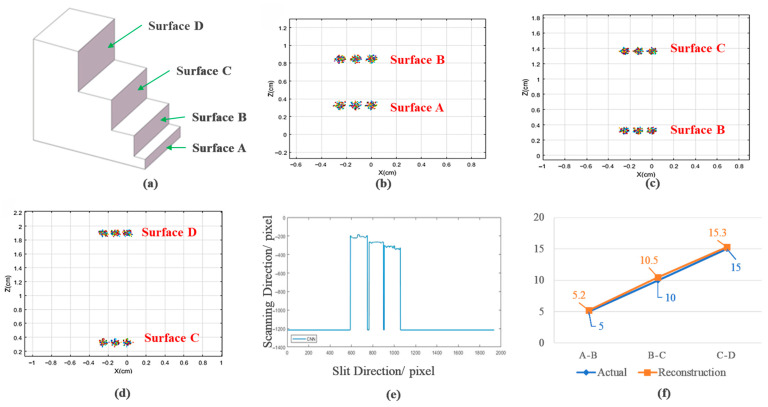
(**a**) Simulated model of the stepped target; (**b**–**d**) simulated data acquisition imaging results for surface pairs A–B, B–C, and C–D; (**e**) simulated imaging results of distance reconstruction; (**f**) error comparison between simulated and reconstructed data.

**Figure 5 sensors-25-05348-f005:**
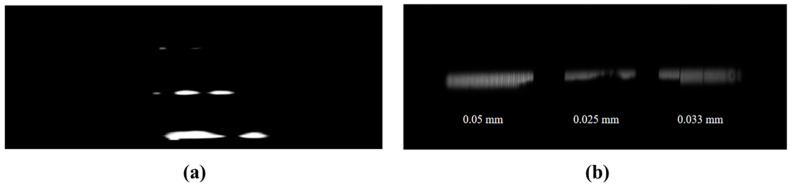
(**a**) Streak images acquired under the third-stage scanning circuit configuration; (**b**) graticule area characterization in static testing.

**Figure 6 sensors-25-05348-f006:**
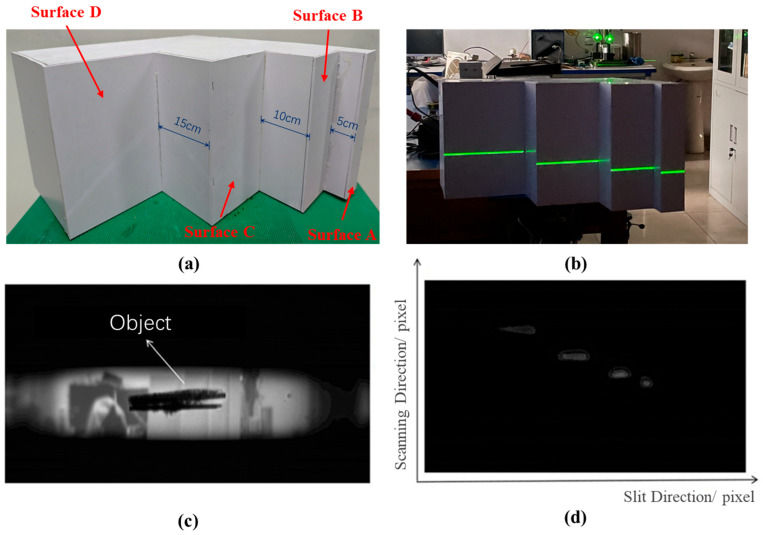
(**a**) Photograph of the target object; (**b**) the pulsed laser beam expanded through a beam collimator and illuminates the target object; (**c**) stepped target geometry; (**d**) reconstructed target image.

**Figure 7 sensors-25-05348-f007:**
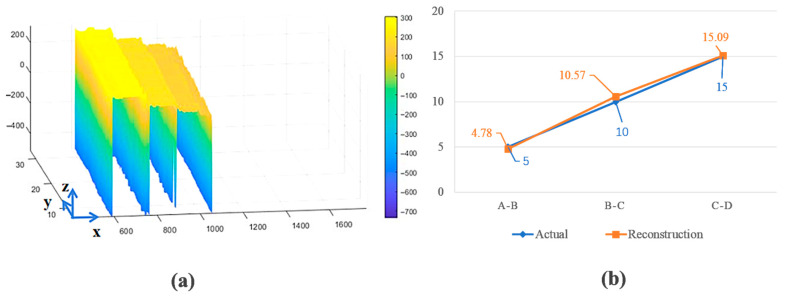
(**a**) Neural network-based 3D reconstruction; (**b**) reconstructed vs. actual distance errors.

**Table 1 sensors-25-05348-t001:** Scanning circuit speed range parameters.

Speed Range	Full-Frame Deflection Voltage	Full-Frame Scanning Duration	Intrinsic Temporal Delay	Nonlinearity
S1	900 V	4.0 ns	41.0 ns	13.28%
S2	900 V	8.4 ns	50.0 ns	7.18%
S3	900 V	20.4 ns	71.6 ns	4.2%
S4	900 V	44.0 ns	123.0 ns	2.7%

## Data Availability

The data are contained within this article.

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
