# Peer review of "Streak Tube-Based LiDAR for 3D Imaging"

_sensors, 2025, doi:10.3390/s25175348_

Round 1

Reviewer 1 Report

Comments and Suggestions for Authors

This paper presents theoretical and experimental investigations of a streak tube imaging LiDAR. The authors establish a theoretical model of the STIL system with numerical simulations predicting limits of temporal and spatial resolutions of the system. Experimental results are provided to validate the capability of the system to realize high frame-rate, high-resolution 3D imaging. My comments and suggestions are as follows:

  1. The summary of the developments of streak tube imaging should include more results from more countries.
  2. A flow diagram is recommended to insert at section 2.
  3. In Fig. 3, (b) should contain the results from more steps corresponding to the texts. In the texts following Fig. 3, some explanations should be added to support Fig. 3 (c) and (d).
  4. In the experiment section, photographs of experimental scene should be added. Also, photographs and information of targets are necessary to be provided as the ground truth.
  5. Some typos are to be corrected, like “SAPDs” should be “SPADs”.

Reviewer 2 Report

Comments and Suggestions for Authors

This manuscript presents the development and demonstration of a streak tube imaging LiDAR (STIL) system capable of high temporal (~14.5 ps experimental) and spatial (up to 22.8 lp/mm) resolution for 3D imaging. A detailed theoretical model is supported by simulation results and experimental validation using a stepped target under both static and dynamic conditions. A key innovation of the work is the integration of a neural network-based 3D reconstruction algorithm that significantly improves depth reconstruction accuracy—reducing errors to 3.57% compared to traditional peak detection approaches.

The manuscript is well-written, technically sound, and demonstrates a strong combination of theoretical modeling, experimental implementation, and algorithmic innovation. The STIL system shows great promise for applications demanding ultrafast, high-resolution 3D imaging.

The topic is original and highly relevant to the fields of ultrafast photonics, LiDAR technology, and time-resolved imaging based on the following few observations throughout the manuscript.

The paper addresses a specific gap in the domain of ultrafast 3D imaging by:

  • Using streak tube cameras (typically used in inertial confinement fusion or x-ray diagnostics) in LiDAR applications.
  • Combining hardware performance with a machine learning-based reconstruction framework to overcome accuracy limitations of peak detection in noisy environments.

This approach stands out because few studies successfully integrate picosecond-scale hardware with neural network-based processing for dynamic 3D imaging.

This paper adds the following key contributions:

  • A detailed simulation and experimental calibration of a STIL system with ~14.5 ps experimental temporal resolution and up to 22.8 lp/mm spatial resolution.
  • A neural network-based 3D reconstruction framework that reduces distance reconstruction errors from ~10% (traditional methods) to 3.57%.
  • A systematic error analysis and performance comparison across simulated and experimental data, including dynamic target imaging.

These additions strengthen the state-of-the-art in high-speed 3D imaging, offering a viable architecture for sub-nanosecond depth sensing that could benefit high-speed metrology, remote sensing, and defense imaging.

 Figures:

    • The schematic diagrams (e.g., Figures 1 and 3) are well-labeled and informative.
    • The raw streak images (Figure 5) and reconstruction comparison (Figure 6) are strong visual validations of the approach.
    • However, some figures could benefit from higher resolution and zoom-in views (e.g., step transitions in Figure 6a).

Tables:

    • Table 1 presents essential scan parameters. It would be helpful to add a column for "Spatial Resolution per Range" for quick cross-comparison.

 All in all this is a great piece of work by the authors and I recommend it for publication provided authors and considering answering following questions.

 Questions for the Authors:

  1. Scalability and Practical Deployment:
    • Could the authors comment on the scalability of the STIL system for practical or commercial applications? Specifically, what are the anticipated challenges related to system integration, cost, or real-time processing for use in industrial or defense settings?
  2. Applicability Beyond LiDAR – e.g., OCT:
    • While not a direct comparison, the streak camera-based framework appears to share conceptual similarities with ultrafast time-domain techniques used in Optical Coherence Tomography (OCT). Could the authors speculate on whether this architecture might be adapted or inspire new OCT implementations, especially for applications requiring extended depth or lateral resolution?
  3. Shorter Laser Pulses and Temporal Resolution:
    • The system demonstrates a temporal resolution of 14.5 ps experimentally. Have the authors considered exploring the use of even shorter laser pulses (e.g., femtosecond regime) to further enhance resolution or ToF sensitivity? If so, what are the anticipated trade-offs in terms of signal-to-noise ratio or system complexity?

 Minor Suggestions:

  1. Spatial Resolution Presentation – MTF Plot:
    • The paper mentions spatial resolution results both in simulation and experiment. Including a Modulation Transfer Function (MTF) plot would provide a more rigorous and standardized way to characterize spatial resolution and system performance across the field of view. This would be especially useful to understand performance degradation near the edges (as noted in the grating test image).
  2. Dynamic Imaging Performance:
    • While the paper discusses dynamic push-broom scanning, it would be helpful to briefly mention any limitations in terms of target motion speed, synchronization accuracy, or required laser repetition rate for real-world 3D imaging scenarios.
  3. Error Sources and Uncertainty:
    • The reconstruction error is quantified well, but could the authors elaborate on the dominant sources of uncertainty—whether they stem from timing jitter, deflection nonlinearity, laser pulse width, or neural network generalization? A breakdown of error contributions would help the community assess the robustness of this approach.

This work presents a strong and novel contribution to the field of ultrafast 3D imaging. The integration of a streak camera with neural network-based depth reconstruction is both timely and technically significant. The paper should be accepted with minor revisions to address the above comments and further strengthen the clarity and utility of the manuscript.
